# Comparison of MAFLD and NAFLD Characteristics in Children

**DOI:** 10.3390/children10030560

**Published:** 2023-03-16

**Authors:** Yunfei Xing, Jiangao Fan, Hai-Jun Wang, Hui Wang

**Affiliations:** 1Department of Maternal and Child Health, School of Public Health, Peking University, Beijing 100191, China; 2Department of Gastroenterology, Xinhua Hospital, Shanghai Jiao Tong University School of Medicine, Shanghai 200092, China

**Keywords:** nonalcoholic fatty liver disease, metabolic-associated fatty liver disease, NHANES, CPOOA, hepatic steatosis

## Abstract

Background & aims: An international panel proposed a diagnostic framework for metabolic-associated fatty liver disease (MAFLD) in children. The aim was to compare the clinical features of MAFLD and nonalcoholic fatty liver disease (NAFLD) in children. Methods: The characteristic differences between NAFLD and MAFLD in children were compared with the National Health and Nutrition Examination Survey (NHANES) 2017–2018 in the U.S. and the Comprehensive Prevention Project for Overweight and Obese Adolescents (CPOOA) study in China. Results: In NHANES 2017–2018, regardless of which criteria were implemented, participants with hepatic steatosis were more likely to have higher BMI z-scores, a higher prevalence of hypertension or higher metabolic indices and higher non-invasive liver fibrosis scores (all *p* < 0.05). The cases diagnosed by those two definitions had a similarity of over 75%. More obese children were diagnosed with MAFLD than NAFLD (*p* < 0.001). However, approximately 19% of children with NAFLD present with normal weight and fasting glucose levels and cannot be diagnosed with MAFLD. The CPOOA study excluded viral infected liver disease and certain kinds of congenital causes of liver steatosis patients, resulting in children with NAFLD being identical with MAFLD children. Conclusions: Most clinical features were similar between children with MAFLD and children with NAFLD, and more than 75% of children with NAFLD can also be diagnosed with MAFLD. However, approximately 19% of children with NAFLD cannot be categorized as MAFLD. Therefore, to gain greater benefits from renaming NAFLD to MAFLD in pediatrics, the prevalence of different causes of hepatic steatosis in children needs to be understood.

## 1. Introduction

Nonalcoholic fatty liver disease (NAFLD) is a series of diseases characterized by excessive deposition of fat in hepatocytes, including simple fatty liver disease, nonalcoholic steatohepatitis and cirrhosis [1]. It is the most common cause of chronic liver disease in many countries and affects nearly 10% of the general pediatric population [2,3]. The prevalence of NAFLD is closely associated with obesity. A meta-analysis of nine general population studies estimated that the prevalence of NAFLD in children aged 1–19 years was 2.3% in normal-weight children, 12.5% in overweight individuals and 36.1% in obese individuals [4]. A recent meta-analysis conducted in Chinese children indicated that the prevalence of NAFLD was 6.3% among the general pediatric population and 40.4% in overweight/obese children [5]. However, the definition of NAFLD is problematic for children, as alcohol consumption is generally not an issue, since most countries have issued a prohibition for children under 18 years of age.

Obesity and metabolic dysfunction are important clinical features of NAFLD; therefore, an international panel proposed the name ‘metabolic-associated fatty liver disease’ (MAFLD) to replace NAFLD in 2020 for adults [6]. Subsequently, an age-specific definition of pediatric MAFLD was released [7]. Compared to the definition of NAFLD, the criteria of MAFLD do not need to exclude other causes of steatosis, such as Wilson’s disease; rather, a requirement is the presence of abnormal metabolic features, including metabolic syndrome, prediabetes or overweight/obesity.

Although some studies have investigated the prevalence of pediatric MAFLD [8,9], the differences in characteristics between MAFLD and NAFLD in children have not been thoroughly explored. Therefore, we used pediatric data from the 2017–2018 National Health and Nutrition Examination Survey (NHANES) of the U.S. and the Comprehensive Prevention Project for Overweight and Obese Adolescents (CPOOA) study of China to compare the clinical features between NAFLD and MAFLD.

## 2. Materials and Methods

### 2.1. Study Design and Population Selection

The first study population was extracted from the NHANES 2017–2018, which was conducted by the National Center for Health Statistics to assess the health and nutritional status of adults and children in the United States. NHANES is a cross-sectional survey program combining interviews and physical examinations, and its complex, multistage, probability sampling design makes the population highly representative. More detailed information about NHANES can be found on the website (https://wwwn.cdc.gov/nchs/nhanes/, accessed on 31 July 2022). In total, 1051 participants aged 12–18 years were recruited in the NHANES 2017–2018. Participants without enough information to be categorized as NAFLD or MAFLD were removed. Specific exclusion criteria were lack of complete vibration-controlled transient elastography (VCTE) (n = 185) or complete absence of important laboratory covariates (n = 72): fasting plasma glucose (FPG), glycosylated hemoglobin (HbA1c), total cholesterol (TC), total triglyceride (TG) and high-density lipoprotein cholesterol (HDL-C); for children aged >15 years, C-reactive protein was also included. After exclusion, 794 participants were included in our study (Figure 1).

The second study population was 1093 Chinese children aged 7–18 years who were extracted from the CPOOA study, as previously described [10]. The CPOOA study was carried out in 2 primary schools and 3 middle schools. Based on their medical history, participants with alcohol use or other liver-damaging factors besides NAFLD, including a history of diseases or drugs (including herbal medicines), infectious hepatic disease (hepatitis B virus [HBV] and hepatitis C virus [HCV]), autoimmune liver disease, Wilson’s disease and α1-antitrypsin (A1AT)-deficiency liver diseases, hepatic malignancies, biliary tract disease and any cardiovascular disease, were excluded from the CPOOA study [10]. Ultimately, ultrasound examination was performed on 1026 children with 1 trained doctor. The CPOOA study was approved by the ethics committee of the Peking University Health Science Center (IRB00001052-06084). All participants and their parents signed informed consent.

### 2.2. Definition of NAFLD and MAFLD

#### 2.2.1. Definition of Hepatic Steatosis

The diagnosis of hepatic steatosis in the NHANES 2017–2018 and the CPOOA was based on VCTE and abdominal ultrasonography, respectively. The criteria in NHANES 2017–2018 were based on two indicators of VCTE: median controlled attenuation parameter (CAP ≥ 248 dBm) or median liver stiffness measurement (≥7.4 kPa) [11]. In CPOOA, the diagnosis of hepatic steatosis requires at least two of the following conditions based on abdominal ultrasonography: diffusely increased echogenicity (‘bright’) liver with liver echogenicity greater than kidney or spleen, vascular blurring and deep attenuation of the ultrasound signal.

#### 2.2.2. Definition of NAFLD

Based on the presence of hepatic steatosis, the diagnosis of NAFLD in Chinese children excluded other liver-damaging factors identified in the medical history (self-report). In NHANES 2017–2018, alcohol consumption, congenital disorders or drug use cannot be excluded due to data limitations. In the present study, we only excluded participants who had positive hepatitis B/C infection or without hepatitis tests (n = 16, Figure 1).

#### 2.2.3. Definition of MAFLD

Children were categorized as having MAFLD based on the presence of hepatic steatosis and at least one of the following conditions: excess adiposity (overweight/obese or abdominal obesity), prediabetes or type 2 diabetes mellitus (T2DM), and metabolic abnormalities [7]. Overweight/obese were defined as having BMI z-scores >1 SD for children ≥ 5 and <10 years according to the WHO growth reference, and the diagnosis of abdominal obesity depended on waist circumference > 90th percentile [12]. Prediabetes was diagnosed by 5.6 ≤ FPG ≤ 6.9 mmol/L or HbA1c 5.7% to 6.4%, while T2DM corresponded to FPG > 7.0 mmol/L or HbA1c > 6.5%. For children between 10 and 15 years old, metabolic abnormalities were defined as the presence of at least two of the following conditions: (1) systolic blood pressure >130 mm Hg or diastolic blood pressure > 85 mm Hg; (2) plasma TG > 150 mg/dL; (3) plasma HDL-C < 40 mg/dL; and (4) TG-to-HDL-C ratio > 2.25. For children > 15 years, the adult criteria of MAFLD were implemented [13].

#### 2.2.4. Definition of Non-Metabolic Dysfunction NAFLD (Non-MD-NAFLD)

Non-MD-NAFLD referred to participants who met the definition of NAFLD but did not meet the definition of MAFLD.

#### 2.2.5. Anthropometric and Biochemical Measurements

In NHANES 2017–2018, anthropometric and biochemical measurements including waist circumference, BMI z-score, systolic blood pressure, diastolic blood pressure, platelet count, HbA1c, FPG, plasma HDL-C, blood urea nitrogen (BUN), serum creatinine, total bilirubin (TBIL), alanine transaminase (ALT), aspartate aminotransferase (AST), γ-glutamyl transferase (GGT), alkaline phosphatase (ALP), total protein, albumin, TC and TG were described in detail elsewhere [14]. Low-density lipoprotein cholesterol (LDL-C) was calculated using the Friedewald formula [15].

In CPOOA, measurements of height, weight, TC, TG, HDL-C, LDL-C, FPG and ALT were performed according to standard protocols, details of which have been reported previously [16].

### 2.3. Non-Invasive Scores

Four non-invasive scores, including the AST-to-platelet ratio index (APRI) [17], NAFLD fibrosis score (NFS, based on age, BMI, diabetes, ALT, AST, platelet count and albumin) [18], Fibrosis-4 index (FIB-4, based on age, AST, ALT and platelet count) and BARD score (based on BMI, AST/ALT, diabetes score) [19] were used to assess liver fibrosis, with cutoff values of 1.5, −1.455, 1.3 and 2, respectively [20]. All of these calculations were performed only in the NHANES 2017–2018.

### 2.4. Statistical Analyses

We performed all analyses using RStudio 2022.02.0. Given the multistage design of the NHANES study, we applied appropriate weights in all analyses based on NHANES data. Continuous variables were represented as weighted means ± SEs and categorical variables as weighted percentages. We used the design-adjusted Rao–Scott χ^2^ test and *t*-test to analyze the differences between categorical variables and continuous variables, respectively. For the analysis based on CPOOA data, continuous variables were represented as means ± SEs and categorical variables as percentages. Statistical significance was defined as a two-tailed *p*-value < 0.05.

## 3. Results

Among 794 participants aged 12–18 years in NHANES 2017–2018, 51.2% were males, with a mean age of 15.03 years and a mean BMI z-score of 1.15. The participants were mainly non-Hispanic white (52.0%). T2DM was identified in 0.6% of participants. The prevalence of MAFLD and NAFLD was 20.7% and 26.4%, respectively. In comparison with children without NAFLD or MAFLD, participants with NAFLD or MAFLD had higher BMI z-scores, higher prevalence of hypertension, higher levels of platelets, higher metabolic indices (FPG, HOMA-IR, HOMA-β, HbA1c, unfavorable serum lipid levels and elevated liver enzymes) and higher non-invasive liver fibrosis scores (all *p* < 0.05) (Table 1). Additionally, most participants with NAFLD were categorized as MAFLD due to overweight/obesity and prediabetes or T2DM. In NHANES 2017–2018, the percentages of overweight/obesity in NAFLD and MAFLD children were 78.0% and 98.0%, respectively (Figure 2). Prediabetes or T2DM is another dominant factor, accounting for 35.2% and 30.7% of children with MAFLD and NAFLD, respectively. Moreover, the proportions of MAFLD and NAFLD participants with at least one of the characteristics (overweight/obesity, prediabetes or T2DM, and metabolic dysregulation) were 100.0% and 82.3% respectively.

Comparisons among NAFLD, MAFLD and non-MD-NAFLD are presented in Table 2. Participants with MAFLD had higher BMI z-scores and lower levels of ALT, TG and HDL-C than those with NAFLD (*p* < 0.001). The participants with non-MD-NAFLD had the lowest mean BMI z-scores (both *p* < 0.001), highest level of HDL-C (both *p* < 0.001), lowest levels of ALT, GGT, HOMA-IR and TG (all *p* < 0.05) and highest FIB-4 score (both *p* < 0.05) compared to the other two groups. In addition, 184 participants had both MAFLD and NAFLD (Figure 3). Forty-one participants met the definition of non-MD-NAFLD, accounting for 18.8% of the participants with NAFLD. Three participants met the definition of MAFLD but not NAFLD, as they lacked information for the diagnosis of hepatitis B/C.

In the CPOOA data, all NAFLD children fulfilled the definition of MAFLD. Therefore, children with NAFLD and children with MAFLD were identical populations (Table 3, Figure 4). All participants with NAFLD or MAFLD had at least one of the three characteristics: overweight/obese, prediabetes or T2DM, and metabolic dysregulation (Figure 2).

## 4. Discussion

The present study is the first study to compare the characteristics of children with MAFLD and children with NAFLD in two different populations. Our study mainly found that MAFLD criteria can identify more obese children than NAFLD criteria. As we expected, in the population of overweight and obese children, MAFLD children were identical to NAFLD children when all infectious or congenital disorders related to liver steatosis were excluded, such as viral hepatitis and Wilson’s disease.

Although the diagnosis of MAFLD differs from the criteria of NAFLD, the cases diagnosed by the two methods had a similarity over 75%. The reason for the high similarity is that both MAFLD and NAFLD are related to overweight/obesity. Previous meta-analyses showed that the prevalence of NAFLD was much higher in overweight/obese children, reaching 36.1% globally and 40.4% in Chinese children, respectively [4,5]. In the present study, the prevalence of MAFLD and NAFLD in overweight/obese children was 49.2% and 49.9% in NHANES 2017–2018, respectively, and both were 25.4% in CPOOA, similar to the results of a contemporaneous study [21]. Notably, approximately 80% of all adolescents with fatty liver disease were diagnosed with MAFLD, which is lower than the proportion reported in a recent study by Ciardullo and colleagues [22]. In that study, the percentage of MAFLD in a steatotic population was around 87.7%. A plausible reason might be that the subjects were collected from 2017 to 2020. Second, both MAFLD and NAFLD are related to hyperglycemia. Previous research indicated that hyperglycemia could drive the metabolic-related NAFLD and that genetic NAFLD could drive the metabolic-related hyperglycemia [23,24]. In a cross-sectional study across 12 centers in the U.S., 23.4% of children with NAFLD had prediabetes, and participants with NAFLD who had T2DM were more likely to have metabolic steatohepatitis [25]. Our data indicated that participants with NAFLD or MAFLD had higher levels of FPG and more severe manifestations of insulin resistance, and 30–40% of children with NAFLD or MAFLD were hyperglycemic. Considering the higher prevalence of MAFLD and NAFLD in children with overweight/obesity and hyperglycemia, screening of liver function should be more widely promoted by the state and government in this population to prevent and control the development of fatty liver.

A recent study [26] explored potential etiologies among 900 overweight/obese children with liver steatosis from two tertiary care centers in North America and found that 347 (39%) children were biopsy-diagnosed with NAFLD and 19 (2%) children had other causes, including thyroid disease, celiac disease, A1AT deficiency, hemochromatosis and Hodgkin’s lymphoma. No cases of viral hepatitis, Wilson’s disease or autoimmune hepatitis (AIH) were found [26]. Consistently, viral hepatitis was not found in children with MAFLD in the present study. Ioannou [27] found that the prevalence of HBV in the U.S. population aged 6 years and older and children aged 6–17 years in NHANES 1999– 2008 was 0.3% and 0.03%. Intriguingly, the prevalence of viral infection with HBV was 0.3% among U.S. adults, and none was detected in NHANES 2017–2018. Both hospital and population survey data indicated that only a few children were infected with HBV through vertical transmission, and the prevalence of hepatitis B is much lower in children than in adults, which might be related to widespread vaccination in children [27,28]. Additionally, Schwimmer et al.’s research carried out in the western United States in 2013 revealed that AIH was the most common alternative diagnosis in non-MD-NAFLD children (4%) [29]. In this context, a small portion of overweight/obese children with steatosis may have alternative etiologies, which vary by area or era. Understanding the constituents and proportion of hepatic steatosis among overweight/obese children and clinical features of different etiologies of hepatic steatosis is important for clinical doctors to find the appropriate treatment in a timely manner after lifestyle intervention if MAFLD is adopted in the future. However, we should bear in mind that MAFLD could also be concurrent with a clear cause of steatosis in children, for instance, HCV or congenital-disorder-induced hepatic steatosis.

The diagnosis of NAFLD requires the exclusion of fatty liver caused by infections, dietary causes (alcohol use), medications or genetic/metabolic disorders. In contrast, the diagnosis of MAFLD does not require such exclusions. In the CPOOA study, all children with NAFLD met all the requirements of MAFLD in CPOOA as shown in Figure 4: participants with MAFLD not only covered the vast majority of participants with NAFLD but also included those with fatty liver caused by infections, diet (alcohol use), medications or genetic/metabolic disorders. It can be inferred that in overweight/obese children, the diagnosis of MAFLD is superior to that of NAFLD, as the former allows the presence of a joint diagnosis. In addition, the proportion of congenital or drug-induced hepatic steatosis is relatively small. Using MAFLD for diagnosis and treatment could reduce the burden from patients and doctors. If weight-loss treatment cannot reverse the progress of hepatic steatosis with three months of lifestyle intervention, doctors could launch more investigations on the etiologies of hepatic steatosis.

Intriguingly, in the present study, 41 participants diagnosed with non-MD-NAFLD seemed healthier than those with MAFLD or NAFLD in terms of metabolic characteristics: they had the lowest BMI z-scores, lowest non-invasive scores, lowest levels of liver transaminase, and they were all under-/normal-weight children, diagnosed with so-called “lean NAFLD”. Genetic susceptibility and other metabolic abnormalities unrelated to weight gain play key roles in these children [30]. The most researched genetic variant for NAFLD is patatin-like phospholipase domain-containing 3 (PNPLA3), which can activate the transcription of thermogenic pathway genes in subcutaneous and brown adipose tissues in TghPNPLA3-I148M mice [31], indicating that genetic NAFLD is highly prone to progress to “lean NAFLD”. Another missense variant, the TM6SF2 rs58542926 (T) allele, could protect against diet-associated obesity [32]. In addition, complex factors including altered energy balance, gut microbiota dysbiosis and insulin resistance all contribute to the development of “lean NAFLD” [33,34,35]. Since these children have a distinct metabolomic profile, the levels of phosphatidylcholine, tyrosine and valine were different from those in the obese group [36]. A study based on NHANES showed that the prevalence of NAFLD among lean children during 2005 to 2014 cycles was 8% [37]. In the present study, the prevalence of NAFLD and MAFLD among lean children in NHANES 2017–2018 was 11.8% and 1.0%, respectively. Although the participants with “lean NAFLD” did not initially show significant metabolic abnormalities, they should not be ignored on account of all possible outcomes of the disease in the long term [38]. In addition, when replacing the diagnosis of NAFLD with MAFLD, these participants with non-MD-NAFLD, accounting for approximately 10–20% of NAFLD, would be missed.

The present study is the first study to compare the characteristics between MAFLD and NAFLD in two different populations. However, several drawbacks need to be mentioned. First, since the data on alcohol use were restrictive for individuals under 18 years old in NHANES 2017–2018, we may have included those who drank excessively. However, we reviewed the summary data from NHANES 2017–2018 for adolescents aged 18 years; only 4 participants drank too much, and none of them was diagnosed with fatty liver. For children aged 12–17 years, only 12 children may have had excessive alcohol use, meaning that alcohol consumption had little effect on the results. Second, we cannot identify other causes of hepatic steatosis, which might increase the disparities between non-MD-NAFLD and MAFLD. Third, VCTE and ultrasound were used to diagnose hepatic steatosis in the present study. Compared with the gold standard liver biopsy, the accuracy of VCTE and ultrasound is limited. However, due to shortcomings such as sampling errors, severe complications and the high cost involved in liver biopsy, non-invasive diagnostic methods such as VCTE or ultrasound are the main detection methods for clinical applications, particularly in children. Fourth, children with diseases such as congenital, viral and drug-induced hepatic steatosis were excluded from the Chinese study by self-report. However, misclassification may occur if the participant had never had the relevant examination at the hospital or if the participant’s recall was biased.

## 5. Conclusions

In conclusion, this study shows that most characteristics were similar between MAFLD and NAFLD. However, approximately 19% of children with NAFLD do not have MAFLD, and this proportion fluctuates with steatosis induced by other causes, such as HCV infection and congenital disorders. Thus, to gain greater benefits from renaming NAFLD to MAFLD in pediatrics, doctors need to understand the prevalence of different causes of hepatic steatosis among children first, and then the appropriate treatment can be delivered to the patients in a timely manner, especially in cases where hepatic steatosis was not alleviated when a lifestyle intervention was carried out and weight loss was achieved. In addition, the presumed causes of hepatic steatosis in children should have a rank list for each area or country, which could help pediatricians treat hepatic steatosis among children in an appropriate and timely way.

## Figures and Tables

**Figure 1 children-10-00560-f001:**
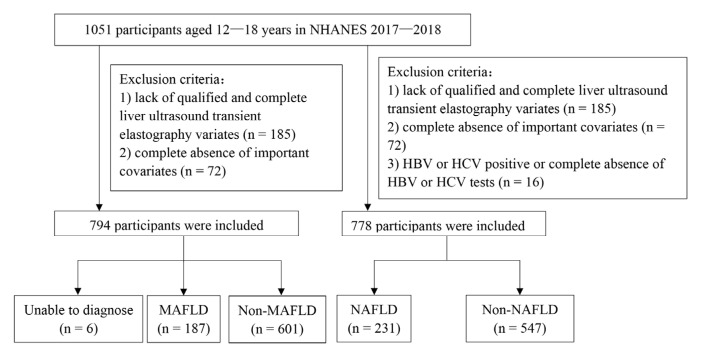
Flow chart of case selection in NHANES 2017–2018. The diagnosis of metabolic-associated fatty liver disease (MAFLD) is shown on the left. Participants with a positive diagnosis of hepatic steatosis who could not be diagnosed with MAFLD or non-MAFLD due to the omission of other indicators were classified as “Unable to diagnose”. The diagnosis of nonalcoholic fatty liver disease (NAFLD) is shown on the right. HBV, hepatitis B virus; HCV, hepatitis C virus.

**Figure 2 children-10-00560-f002:**
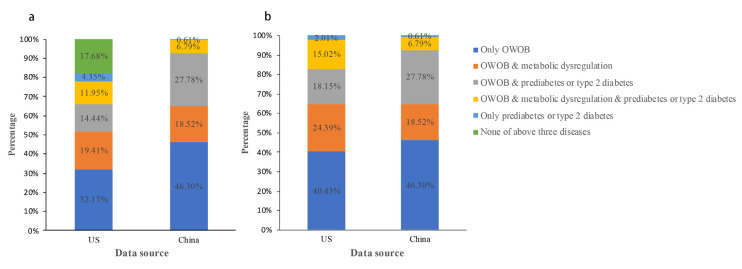
Characteristic composition of NAFLD participants (**a**) and MAFLD participants (**b**) in NHANES 2017–2018 and CPOOA. US and China correspond to the 2017–2018 National Health and Nutrition Examination Survey (NHANES) and the Comprehensive Prevention Project for Overweight and Obese Adolescents (CPOOA), respectively. (**a**,**b**) show the etiological composition of participants with metabolic-associated fatty liver disease (MAFLD) and participants with nonalcoholic fatty liver disease (NAFLD) in the two studies, respectively. Dark blue indicates the presence of the factor OWOB only, orange indicates the presence of both OWOB and metabolic dysregulation, grey indicates the presence of both OWOB and prediabetes or type 2 diabetes, yellow indicates the presence of all three factors: OWOB, metabolic dysregulation and prediabetes or type 2 diabetes, and light blue indicates the presence of only one factor: prediabetes or type 2 diabetes. Green means none of the above three factors were present. OWOB, overweight/obesity.

**Figure 3 children-10-00560-f003:**
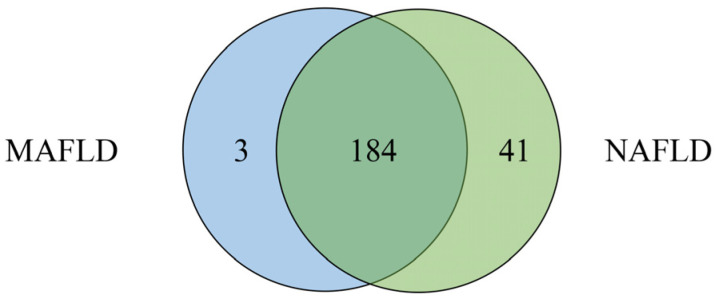
Composition of MAFLD and NAFLD in NHANES 2017–2018. The blue circle indicates participants with metabolic-associated fatty liver disease (MAFLD), and the green circle indicates participants with nonalcoholic fatty liver disease (NAFLD); the middle overlapping part indicates participants with both MAFLD and NAFLD.

**Figure 4 children-10-00560-f004:**
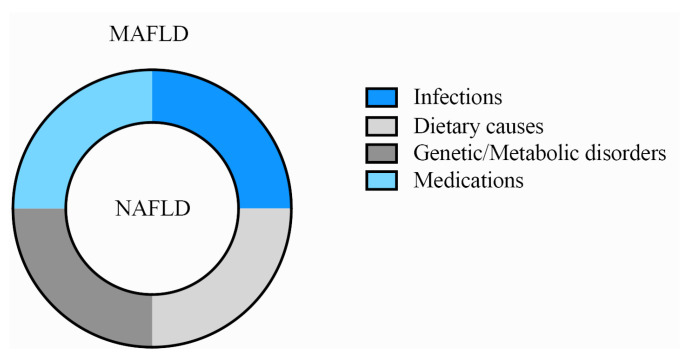
The relationship between MAFLD and NAFLD (wheel model). The inner circle represents nonalcoholic fatty liver disease (NAFLD), and the outer circle represents metabolic-associated fatty liver disease (MAFLD). The four colors indicate infections, dietary causes, medications and genetic/metabolic disorders.

**Table 1 children-10-00560-t001:** Baseline characteristics of subjects in NHANES 2017–2018 by VCTE.

Variables	Based on the MAFLD Criteria (N = 788)					Based on the NAFLD Criteria (N = 778)				
	Non-MAFLD	N	MAFLD	N	*p*	Non-NAFLD	N	NAFLD	N	*p*
Age (years)	14.98 ± 1.98	601	15.21 ± 0.20	187	0.349	14.97 ± 0.09	547	15.25 ± 0.15	231	0.189
Male (%)	49.3	302	58.4	107	0.057	49.2	274	57.1	130	0.156
Race					0.010					0.013
Non-Hispanic White	55.3	201	40.3	49	/	56.2	189	41.8	59	/
Non-Hispanic Black	11.6	127	12.9	35	/	11.2	110	14.1	48	/
Hispanic	20.5	143	34.8	61	/	20.8	133	31.7	70	/
Others	12.7	130	12.1	42	/	11.9	115	12.4	54	/
BMI-z	0.53 ± 0.06	596	3.50 ± 0.11	187	<0.001	0.55 ± 0.06	542	2.80 ± 0.16	231	<0.001
T2DM (%)	0.4	3	1.3	2	0.162	0.4	3	1.1	2	0.307
Hypertension (%)	0.7	6	8.4	15	<0.001	0.8	6	5.9	14	<0.001
Hepatitis B	0	0	0	0	/	/	/	/	/	/
Platelets (×10^9^/L)	259.87 ± 3.29	601	280.19 ± 4.68	187	0.002	259.29 ± 3.39	547	276.83 ± 3.68	231	0.002
BUN (mmol/L)	4.31 ± 0.05	571	4.22 ± 0.09	182	0.214	4.31 ± 0.04	530	4.21 ± 0.09	228	0.175
Creatinine (μmol/L)	62.14 ± 0.51	571	62.64 ± 1.15	182	0.692	62.20 ± 0.50	530	62.55 ± 1.11	228	0.778
TBIL (μmol/L)	8.31 ± 0.36	571	7.05 ± 0.35	182	0.031	8.25 ± 0.37	530	7.49 ± 0.47	228	0.250
ALT (U/L)	14.12 ± 0.49	571	23.47 ± 1.27	182	<0.001	14.15 ± 0.48	530	21.46 ± 1.30	228	<0.001
AST (U/L)	19.52 ± 0.34	568	22.09 ± 0.79	182	0.016	19.48 ± 0.34	527	21.66 ± 0.71	228	0.019
GGT (U/L)	12.85 ± 0.27	571	18.87 ± 1.14	182	<0.001	12.89 ± 0.29	530	17.65 ± 1.05	228	<0.001
ALP (U/L)	160.04 ± 5.23	572	150.56 ± 9.45	182	0.438	158.22 ± 6.14	531	156.91 ± 9.61	228	0.923
Total protein (g/dL)	72.11 ± 0.26	571	72.79 ± 0.36	182	0.064	72.10 ± 0.28	530	72.67 ± 0.36	228	0.152
Albumin (g/L)	43.22 ± 0.20	572	42.04 ± 0.33	182	<0.001	43.17 ± 0.21	531	42.39 ± 0.35	228	0.035
HbA1c (%)	5.24 ± 0.01	600	5.29 ± 0.03	187	0.040	5.24 ± 0.01	546	5.28 ± 0.02	231	0.025
HOMA-IR	2.50 ± 0.16	267	5.23 ± 0.41	90	<0.001	2.51 ± 0.16	249	4.73 ± 0.40	109	<0.001
HOMA-β	109.35 ± 8.88	267	205.86 ± 13.61	90	<0.001	109.65 ± 8.99	249	188.24 ± 13.36	109	<0.001
FPG (mmol/L)	5.41 ± 0.04	280	5.49 ± 0.04	91	0.007	5.42 ± 0.04	255	5.48 ± 0.04	110	0.016
TC (mmol/L)	3.98 ± 0.05	578	4.09 ± 0.07	183	0.193	3.98 ± 0.05	537	4.08 ± 0.05	229	0.183
TG (mmol/L)	0.96 ± 0.04	571	1.34 ± 0.08	182	<0.001	0.97 ± 0.04	530	1.24 ± 0.08	228	0.001
LDL-C (mmol/L)	2.22 ± 0.07	267	2.41 ± 0.06	91	0.018	2.23 ± 0.06	249	2.34 ± 0.06	110	0.160
HDL-C (mmol/L)	1.40 ± 0.02	578	1.19 ± 0.02	183	<0.001	1.39 ± 0.03	537	1.25 ± 0.02	229	0.001
APRI score	0.25 ± 0.01	568	0.27 ± 0.01	182	0.310	0.25 ± 0.01	527	0.27 ± 0.01	228	0.282
APRI score > 1.5, n (%)	0.04	1	0	0	0.631	0.05	1	0	0	0.564
NFS score	−3.60 ± 0.52	563	−3.09 ± 0.09	182	<0.001	−3.59 ± 0.05	522	−3.23 ± 0.11	228	0.014
NFS score > −1.455, n (%)	0.02	14	0.06	14	0.001	1.8	13	6.6	15	0.031
FIB4 score	0.32 ± 0.01	568	0.27 ± 0.01	182	<0.001	0.32 ± 0.01	527	0.29 ± 0.01	228	0.027
FIB4 score > 1.3, n (%)	0.04	1	0	0	0.631	0.05	1	0	0	0.564
BARD score	2.01 ± 0.02	563	2.29 ± 0.05	182	<0.001	2.02 ± 0.02	522	2.22 ± 0.04	228	<0.001
BARD score > 2, n (%)	5.53	44	53.13	90	<0.001	5.91	44	42.0	90	<0.001

Continuous values are means ± SEs and categorical variables are numbers and weighted proportions. We used a *t*-test and the Rao–Scott chi-squared test to compare the differences between groups. VCTE: vibration-controlled transient elastography; MAFLD: metabolic-associated fatty liver disease; NAFLD: nonalcoholic fatty liver disease; NHANES: National Health and Nutrition Examination Survey; T2DM: type 2 diabetes mellitus.

**Table 2 children-10-00560-t002:** Comparison of clinical parameters according to presence of MAFLD, NAFLD and non-MD-NAFLD group by VCTE in NHANES 2017–2018.

Variables	NAFLD (1)	N	MAFLD (2)	N	Non-MD-NAFLD (3)	N	*p*		
							1 vs. 2	1 vs. 3	2 vs. 3
N		231		187		41			
Age (years)	15.25 ± 0.15	231	15.21 ± 0.20	187	15.13 ± 0.23	41	0.784	0.812	0.895
Male (%)	57.1	130	58.4	107	53.5	18	0.566	0.709	0.677
BMI-z	2.80 ± 0.16	231	3.50 ± 0.11	187	0.15 ± 0.06	41	<0.001	<0.001	<0.001
T2DM (%)	1.1	2	1.3	2	0	0	<0.001	0.503	0.442
Hypertension (%)	5.9	14	8.4	15	0	0	0.015	0.129	0.134
Platelets (×10^9^/L)	276.83 ± 3.68	231	280.19 ± 4.68	187	268.20 ± 6.41	41	0.110	0.195	0.165
BUN (mmol/L)	4.21 ± 0.09	228	4.22 ± 0.09	182	4.21 ± 0.12	41	0.893	0.993	0.984
Creatinine (μmol/L)	62.55 ± 1.11	228	62.64 ± 1.15	182	61.31 ± 1.44	41	0.860	0.428	0.517
TBIL (μmol/L)	7.49 ± 0.47	228	7.05 ± 0.35	182	9.06 ± 0.94	41	0.109	0.200	0.176
ALT (U/L)	21.46 ± 1.30	228	23.47 ± 1.27	182	13.68 ± 0.67	41	0.003	<0.001	<0.001
AST (U/L)	21.66 ± 0.71	228	22.09 ± 0.79	182	20.13 ± 0.72	41	0.118	0.052	0.063
GGT (IU/L)	17.65 ± 1.05	228	18.87 ± 1.14	182	12.32 ± 0.50	41	0.009	<0.001	<0.001
ALP (IU/L)	156.91 ± 9.61	228	150.56 ± 9.45	182	186.51 ± 13.39	41	0.407	0.312	0.329
Total protein (g/dL)	72.67 ± 0.36	228	72.79 ± 0.36	182	72.16 ± 0.40	41	0.307	0.348	0.329
Albumin (g/L)	42.39 ± 0.35	228	42.04 ± 0.33	182	43.86 ± 0.23	41	0.019	<0.001	0.001
HbA1c (%)	5.28 ± 0.02	231	5.29 ± 0.03	187	5.26 ± 0.02	41	0.247	0.641	0.498
HOMA-IR	4.73 ± 0.40	109	5.23 ± 0.41	90	2.37 ± 0.17	18	0.292	<0.001	<0.001
HOMA-β	188.24 ± 13.36	109	205.86 ± 13.61	90	104.44 ± 6.73	18	0.269	<0.001	<0.001
FPG (mmol/L)	5.48 ± 0.04	110	5.49 ± 0.04	91	5.42 ± 0.03	18	0.767	0.285	0.222
TC (mmol/L)	4.08 ± 0.05	229	4.09 ± 0.07	183	3.97 ± 0.08	41	0.762	0.385	0.437
TG (mmol/L)	1.24 ± 0.08	228	1.34 ± 0.08	182	0.78 ± 0.03	41	0.005	<0.001	<0.001
LDL-C (mmol/L)	2.34 ± 0.06	110	2.41 ± 0.06	91	2.03 ± 0.07	18	0.470	0.118	0.062
HDL-C (mmol/L)	1.25 ± 0.02	229	1.19 ± 0.02	183	1.50 ± 0.02	41	<0.001	<0.001	<0.001
APRI score	0.27 ± 0.01	228	0.27 ± 0.01	182	0.26 ± 0.01	41	0.705	0.490	0.537
APRI score > 1.5, n (%)	0	0	0	0	0	0	/	/	/
NFS score	−3.23 ± 0.11	228	−3.09 ± 0.09	182	−3.81 ± 0.14	41	0.019	0.008	0.007
NFS score > −1.455, n (%)	6.6	15	0.06	14	0	0	0.905	0.807	0.823
FIB4 score	0.29 ± 0.01	228	0.27 ± 0.01	182	0.34 ± 0.02	41	0.004	0.009	0.006
FIB4 score > 1.3, n (%)	0	0	0	0	0	0	/	/	/
BARD score	2.22 ± 0.04	228	2.29 ± 0.05	182	1.96 ± 0.03	41	<0.001	0.002	<0.001
BARD score > 2, n (%)	42.0	90	53.13	90	0	0	<0.001	<0.001	<0.001

Continuous values are means ± SEs and categorical variables are numbers and weighted proportions. We used a *t*-test and the Rao–Scott chi-squared test to compare the differences between groups. VCTE: vibration-controlled transient elastography; MAFLD: metabolic-associated fatty liver disease; NAFLD: nonalcoholic fatty liver disease; NHANES: National Health and Nutrition Examination Survey; T2DM: type 2 diabetes mellitus.

**Table 3 children-10-00560-t003:** Comparison of clinical parameters of participants diagnosed with MAFLD and NAFLD in CPOOA by ultrasound.

Variables	Based on the MAFLD/NAFLD Criteria (N = 1026)		
	Non-MAFLD/NAFLD (N = 864)	MAFLD/NAFLD (N = 162)	*p*
Age (years)	11.44 ± 0.11	11.81 ± 0.17	0.128
Male (%)	53.1	71.0	<0.001
BMI-z	0.97 ± 0.04	3.13 ± 0.11	<0.001
T2DM	0.2	1.2	0.120
Hypertension	6.1	17.9	<0.001
ALT (U/L)	12.95 ± 0.41	26.90 ± 2.41	<0.001
FPG (mmol/L)	5.34 ± 0.01	5.48 ± 0.03	<0.001
TC (mmol/L)	4.09 ± 0.01	4.23 ± 0.06	0.021
TG (mmol/L)	0.87 ± 0.12	1.17 ± 0.04	<0.001
LDL-C (mmol/L)	2.17 ± 0.02	2.50 ± 0.05	<0.001
HDL-C (mmol/L)	1.54 ± 0.01	1.36 ± 0.02	<0.001

Continuous values are means ± SEs, and categorical variables are numbers and weighted proportions. We used a *t*-test and the Rao–Scott chi-squared test to compare the differences between groups. MAFLD: metabolic-associated fatty liver disease; NAFLD: nonalcoholic fatty liver disease; T2DM: type 2 diabetes mellitus; CPOOA: Comprehensive Prevention Project for Overweight and Obese Adolescents.

## Data Availability

The NHANES 2017–2018 is a public data set and can be accessed.

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
