# Peer review of "Comparison of MAFLD and NAFLD Characteristics in Children"

_children, 2023, doi:10.3390/children10030560_

Round 1

Reviewer 1 Report

Here, the authors compared the clinical features of MAFLD and NAFLD in pediatric patients from China and the USA, using two clinical datasets (one from each country; each dataset includes more than 1000 patients). Similarities and differences were uncovered, such as MAFLD criteria identified more obese children than NAFLD. The manuscript is well-written and the methods are thoroughly described. Results are clearly presented and discussed. This work should be useful to clinicians diagnosing and treating children with MAFLD, NAFLD, and related disorders.

I have a few minor recommendations:

1. In the introduction, it is mentioned that MAFLD requires the presence of two or more metabolic abnormalities. Although these abnormalities are detailed in the materials and methods section, the authors could also provide a few examples in the introduction.

2. In the legend of figure 1, please include the full names of the HBV and HCV abbreviations used in the figure.

3. In figure 2, what is "OWOB"? Is it "overweight/obesity"? If so, please explain in the respective legend.

4. In figures 3 and 4, the authors could use a more diverse set of colors (not restricted to grayscale).

5. In the discussion, figure 4 should be explained in more detail.

Author Response

Reviewer 1

I have a few minor recommendations:

  1. In the introduction, it is mentioned that MAFLD requires the presence of two or more metabolic abnormalities. Although these abnormalities are detailed in the materials and methods section, the authors could also provide a few examples in the introduction.

Response:Thanks for your suggestions. We already added in the introduction part. See page 2.

  1. In the legend of figure 1, please include the full names of the HBV and HCV abbreviations used in the figure.

Response:Thank you very much, We comprehensively modified the figure legend. See page 2.

  1. In figure 2, what is "OWOB"? Is it "overweight/obesity"? If so, please explain in the respective legend.

Response:We added that information in the corresponding legend.

  1. In figures 3 and 4, the authors could use a more diverse set of colors (not restricted to grayscale).

Response: We changed the color as your suggestion.

  1. In the discussion, figure 4 should be explained in more detail.

Response: We added more explanation in the discussion part. See page 15.

Reviewer 2 Report

In the present manuscript the authors report the results of a cross-sectional study performed in two cohorts of adolescents (from the US and China) aimed at evaluating the degree of concordance between the NAFLD and MAFLD definitions, as well as differences across groups. The topic is certainly timely given the intense debate regarding this new definition in the literature. I have the following comments:

-          English is sub-standards and needs improvement. I suggest seeking help from a native speaker

-          When analyzing the NHANES data, authors have to apply sampling weights to all analyses. There is no description of weights being used in the statistical analysis section. Please use them.

-          A recent study performed in the 2017-2020 NHANES cohort should be mentioned in order to identify the novelty of the present analysis (doi: 10.1002/hep4.1969)

Author Response

 English is sub-standards and needs improvement. I suggest seeking help from a native speaker

 Response: We adopt the Editing service from AJE and comprehensively revised our writing. Thanks for your suggestion.   

-          When analyzing the NHANES data, authors have to apply sampling weights to all analyses. There is no description of weights being used in the statistical analysis section. Please use them.

Response: thanks for your valuable suggestion. We applied sampling weights to all analyses now. We searched the weighting methods online for NHANES and we adopt the WTMEC2YR this weighting method. Since if we used the WTSAF2YR, around 50% data will be lost. Therefore, we chose the WTMEC2YR for weighting. For the statistical analysis, the 'survey' package in R software were used for analysis.  Namely, adjusted Rao-Scott χ2 test and t-test were adopted to analyze the differences be-tween categorical variables and continuous variables. See page 4.

-          A recent study performed in the 2017-2020 NHANES cohort should be mentioned in order to identify the novelty of the present analysis (doi: 10.1002/hep4.1969)

Response: we added several sentences to explain what differences between Ciardullo’s study and our present study. Please see page 15.

Reviewer 3 Report

The authors found that most clinical features are similar between MAFLD and NAFLD in children, but some children have NAFLD and not MAFLD. It is interesting but some points to be revised.

-In line 89 biliary tract disease, and any cardiovascular disease. Please describe these in more detail by listing the specific names of the diseases.

-In Figure 2, please spell out the abbreviation "OWOB".

-It is unnatural that the average age indicated in Table 3 is about 11 years old, while the CPOOA targets children between 12 and 18 years old, and requires an explanation.

-Asians have a relatively abundant population of NAFLD patients, despite the fact that they are clearly leaner than Americans. Can the authors compare the difference in contribution of body weight (BMI) to fatty liver disease in NHANES and CPOOA? 

- The information presented in Table 3 is too little. In CPOOA, can the authors compare liver stiffness indices (including FIB4 or AST/ALT ratio) in non-MAFLD/NAFLD (n=864) and MAFLD/NAFLD (n=162) ?

Author Response

-In line 89 biliary tract disease, and any cardiovascular disease. Please describe these in more detail by listing the specific names of the diseases.

Response: thank you very much for your question. This survey carried out in 2008, during that time, the field worker just asked the primary students whether they have been diagnosed with biliary tract disease or any cardiovascular disease generally, no specific disease was pinpointed.

-In Figure 2, please spell out the abbreviation "OWOB".

Response: thanks. We write the full name of OWOB in the page five.

-It is unnatural that the average age indicated in Table 3 is about 11 years old, while the CPOOA targets children between 12 and 18 years old, and requires an explanation.

Response: thanks for point out this problem. It is our carelessness. Right now, we correct them accordingly. The CPOOA target children between 7~18 years old.

 -Asians have a relatively abundant population of NAFLD patients, despite the fact that they are clearly leaner than Americans. Can the authors compare the difference in contribution of body weight (BMI) to fatty liver disease in NHANES and CPOOA? 

Response: The CPOOA study was a weight interventional study, which mainly enrolled overweight and obese children. In the total population, we had 429 normal weight children and 597 overweight/obese children.  The mean BMI-z score of Chinese children is higher than that of American children, but the mean BMI-Z score of children with MAFLD is lower, and the prevalence of fatty liver disease is also lower than that of American children.

Adjusted for age and gender, logistic regression of BMIz score on fatty liver was performed in two studies, as shown in the above table. The results showed that BMIz score had a greater impact on fatty liver in Chinese children than the children than in American children.

Variables

β

P value

OR

CPOOA

age

-0.231

<0.001

0.794

sex

-0.279

0.223

0.757

BMI-z

1.390

<0.001

4.015

NHANES

age

0.117

0.016

1.124

sex

-0.398

0.034

0.672

BMI-z

0.701

<0.001

2.016

- The information presented in Table 3 is too little. In CPOOA, can the authors compare liver stiffness indices (including FIB4 or AST/ALT ratio) in non-MAFLD/NAFLD (n=864) and MAFLD/NAFLD (n=162)

Response: We are sorry that we cannot conduct this calculation. In the COOPA study, we did not test the serum AST in primary students. Based on the published documents, the AST level did not change so much as ALT in children (doi: 10.3389/fped.2021.629346, doi: 10.3389/fpubh.2022.991393).

Round 2

Reviewer 2 Report

I congratulate the Authors for their thorough revision. I have no further comments.

Reviewer 3 Report

None.